# Electronic and Optical Properties of Twin T-Graphene Co-Doped with Boron and Phosphorus

**DOI:** 10.3390/ma15082876

**Published:** 2022-04-14

**Authors:** Yue Gao, You Xie, Sufang Wang, Shuang Li, Liyong Chen, Jianmin Zhang

**Affiliations:** 1College of Science, Xi’an University of Science and Technology, Xi’an 710054, China; gaoyue19980205@163.com (Y.G.); sufangwang@xust.edu.cn (S.W.); shuangli66@outlook.com (S.L.); liyongchen@xust.edu.cn (L.C.); 2College of Physics and Information Technology, Shaanxi Normal University, Xi’an 710119, China; jmzhang@snnu.edu.cn

**Keywords:** twin T-graphene, co-doping, electronic structure, optical property, first-principles calculation

## Abstract

Twin T-graphene (TTG) is a new two-dimensional carbon allotrope of graphene. Heteroatom co-doping is an effective method for the modulation of the physical and chemical properties of two-dimensional materials. This study explored the structural stability, electronic structures, and optical properties of boron and phosphorus co-doped TTG using first-principles calculations. TTG was doped with B and P atoms (BP) at different positions considering 13 different configurations. Pristine TTG has a band gap of 1.89 eV, and all BP co-doped TTG (TTG/BP) systems remain semiconducting with band gaps that gradually decrease with increasing doping concentration. For a given doping concentration, the TTG/BP-ortho systems had a narrower band gap than the corresponding TTG/BP-para systems. The TTG and TTG/BP systems exhibited significant optical anisotropy. In the infrared region, BP co-doping increased the absorption coefficient, and the reflectance and refractive index increased with increasing doping concentration, except for the vertical component of the TTG/BP-ortho system. In the visible region, the absorption coefficient, reflectance, and refractive index decreased with increasing doping concentration for the vertical component, and the peaks were red-shifted from the near-ultraviolet region to the visible region. In the near-ultraviolet region, the reflectance also decreased with increasing doping concentration. The BP co-doping concentration can regulate the electronic structures and optical properties of the TTG, showing that the BP co-doped TTG has potential for application in nanoelectronics and optoelectronics.

## 1. Introduction

Twin T-graphene (TTG)—a new two-dimensional carbon allotrope with a thickness of two atomic layers [1]—shows sp^2^ hybridization with a 4–16 membered ring structure where two tetragonal rings are stitched together with four carbon atoms. Two-dimensional TTG has excellent thermal stability at room temperature and its bonding pattern does not change even at 2000 K. The TTG sheet is an intrinsic nonmagnetic semiconductor with a band gap of 1.79 eV that is much larger than that of twin graphene (1.0 eV) and monolayer graphene [2,3]. Moreover, the elastic constant of TTG is approximately 177 N·m^−1^, and its carrier mobility is approximately 375 cm^2^·V^−1^·s^−1^, making semiconducting TTG suitable for use in flexible electronic and optoelectronic devices such as field-effect transistors and solar cells.

Bhattacharya et al. reported that nitrogen doped TTG sheet is a bipolar magnetic semiconductor with a spin-flip band gap (1.15 eV) that has good application potential in spintronic devices [1]. Majidi et al. systematically studied the electronic structures of TTG doped with 3d transition metals (TM = Sc, Ti, V, Cr, Mn, Fe, Co, Ni, Cu, and Zn) [4]. They found that TM doping can modulate the electronic properties of TTG, and the doping of different species and concentrations of TM atoms resulted in different electronic properties. TM-doped TTG systems show semiconducting behavior for Sc, Ti, V, Cr, and Zn doping, metallic behavior for Mn, Cu, and Ni doping, and bipolar magnetic semiconducting behavior for Fe and Co doping. For TM-doped TTG systems, the band gap decreased with increasing concentration of the 3d transition metal atom dopants. These results show that both pristine and metal-doped TTG can be used for electronic and spintronic applications. However, no studies have reported doped TTG related to their optical properties; therefore, it is important to explore whether TTG has potential applications in the optical devices.

Doping can modify the magnetic, electronic, transport, and optical properties of two-dimensional materials, particularly for graphene and its allotropes [5,6,7,8,9,10,11,12,13]. In practice, due to the strong thermodynamic driving force, it is difficult to achieve single-atom doping because atoms doped into two-dimensional materials often form clusters [14]. Co-doped graphene was found to be relatively easier to synthesize compared to mono-doped graphene. Moreover, compared to doping of individual elements, the doping of carbon materials with multiple elements is more advantageous for enhancing the physical and chemical properties of the material. Therefore, hetero-elemental co-doping is an effective approach for the modulation of the electronic structure, transport, and catalytic properties of graphene [15,16,17,18,19,20,21,22,23,24,25]. Denis et al. confirmed that co-doping of graphene is much easier to achieve than mono-doping for both 2p elements (B, N, or O) and 3p elements (Al, Si, P or S) [18]. Co-doping is also an excellent strategy for modulating the interaction of lithium with graphene when lithium atoms are adsorbed on heteroatom mono (X) and dual (XY) doped graphene (X = Al, Si, P and S and Y = B, N, and O) [25]. Many experiments on co-doping have been reported. Wang et al. designed two simple methods for the synthesis of unique N, P and N, S co-doped Mo_2_C/C hybrid electrocatalysts as highly active hydrogen evolution reaction catalysts [26]. Li et al. synthesized Bi and S co-doped ZnO samples with different doping ratios [27]. Zhang et al. experimentally studied the Li ion storage mechanism of N and S co-doped V_2_CTx MXene [28]. Based on the previous success of the co-doping strategy for the modification of two-dimensional materials, it is crucial to further explore the properties of co-doped TTG, particularly for TTG co-doping with nonmetallic elements.

Herein, we methodically investigated and analyzed the structural stability, electronic structures, and optical properties of TTG co-doped with nonmetal atoms (B, P) at different doping concentrations. Replacement of C atoms with P atoms can effectively optimize the electronic structure of carbon-based materials [26]. Meanwhile, the doping of 2p B can reduce the structural distortion caused by 3p P into the intrinsic TTG structure. Although it is difficult to control the doping of different atoms at specific sites, such control has been realized experimentally in previous studies. Zhao et al. successfully fabricated N (2p element) and S (3p element) hetero-element co-doped few-layer graphdiyne oxide catalysts via an improved pyrolysis method by providing continuous doping sources [29]. The stereo positions of N and S atoms was well-controlled due to the site-controlled doping of the sp-N atoms in graphdiyne. In [30], selective doping of Mg was conducted on tetrahedral (8a) as well as octahedral (16c) sites in the $\mathrm{Fd}\bar{3}m$ structure. This site-selective doping not only suppresses unfavorable two-phase reactions and stabilizes the LiNi_0_._5_Mn_1_._5_O_4_ structure with respect to structural deformation, but also mitigates the dissolution of Mn during cycling. These successful experiments suggest that fabrication of B and P co-doped TTG is feasible and the positions of the B and P atoms can be controlled during the fabrication process. The theoretical research performed in this study may provide insights for future experiments on heteroatomic co-doping of TTG. In addition, comprehensive understanding of properties of TTG co-doped with the nonmetal atoms and the full exploitation of the entire range of possible optical properties modulated by co-doping concentration can promote the design and development of novel TTG-based optoelectronic devices.

## 2. Methodology

First-principles calculations were carried out using the VASP code [31]. The electron-ion interactions were considered using the highly accurate projector-augmented wave (PAW) potentials [32], and the exchange-correlation energy and potential were evaluated using the Perdew–Burke–Ernzerhof (PBE) formulation of the generalized gradient approximation (GGA) functional [33], producing the correct ground state of the system. To correctly describe the effect of van der Waals (vDW) interactions, we employed the DFT-D3 Grimme scheme [34]. To avoid the interaction between images, a vacuum region of 16 Å was placed perpendicular to the model plane. The plane wave kinetic energy cut-off was set to 450 eV. To optimize the geometry and calculate the electronic structure and optical properties of the model systems, the Monkhorst–Pack method was used to generate a 6 × 6 × 1 k-point sampling grid points for Brillouin zone integration [35]. The total energy and forces were converged to 10^−4^ and 0.01 eV/Å, respectively, for all relaxed geometric structures. To determine the thermodynamic stability of the model structure, the formation energy was defined as:(1)$$E_{form}=[(E\mathrm{TTG}/\mathrm{BP})+n\mu_{C-C}-E\left(\mathrm{TTG}\right)-n\mu_{B-P}]/N$$
where *E*(TTG/BP) and *E*(TTG) are the total energies of the BP co-doped TTG and TTG, respectively; *N* and *n* are the total number of atoms in the supercell and the number of C–C pairs replaced by B–P pairs, and $\mu_{C-C}$ and $\mu_{B-P}$ are the chemical potentials of the C–C and B–P pairs calculated from the graphene and h-BP monolayers, respectively. Charge transfer was calculated quantitatively using Bader charge analysis [36,37].

To investigate the optical properties of the B and P co-doped TTG, we calculated the absorption coefficient, reflectance, and refractive index of the doped system. These optical coefficients are calculated from the frequency-dependent dielectric function. The tensor component of the complex dielectric function can be defined as the sum of the real part ($\varepsilon_{r}$) and the imaginary part ($\varepsilon_{i}$) as follows: $\varepsilon \left(\omega \right)=\varepsilon_{r}\left(\omega \right)+i\varepsilon_{i}\left(\omega \right)$. The imaginary part is determined by summing the empty states according to [38]:(2)$$\varepsilon_{i}\left(\omega \right)=\frac{4\pi^{2}e^{2}}{\Omega }\underset{q\to 0}{\lim}\frac{1}{q^{2}}\underset{c,v,k}{\sum }2w_{k}\delta \left(\varepsilon_{ck}-\varepsilon_{vk}-\omega \right)\times (u_{ck}+e_{\alpha }q|u_{vk})(u_{ck}+e_{\beta }q|u_{vk})$$
where Ω is the volume of the primitive cell, $w_{k}$ is the k-point weight, c and v refer to the conduction and valence bands, respectively, and $\varepsilon_{ck}$, $\varepsilon_{yk}$, and $u_{ck}$, $u_{vk}$ represent the eigenvalues and wavefunctions at the k-point, respectively. The real part of the dielectric function is calculated according to the Kramers–Kronig transformation as follows [38]:(3)$$\varepsilon_{r}\left(\omega \right)=1+\frac{2}{\pi }P\int_{0}^{\infty }\frac{\varepsilon_{i}\left(\omega^{,}\right)\omega^{,}}{{\omega ,}^{2}-\omega^{2}+i\eta }d\omega^{,}$$

Consequently, other important optical properties, such as the optical absorption coefficient (α(ω)), reflectance (r(ω)), and refractive index (n(ω)) can be calculated from the dielectric function as follows [38]:(4)α(ω)=2ω[εr2(ω)−εi2(ω)−εr(ω)]12
(5)$$r\left(\omega \right)={\left|\frac{\sqrt{\varepsilon_{r}\left(\omega \right)+i\varepsilon_{i}\left(\omega \right)}-1}{\sqrt{\varepsilon_{r}\left(\omega \right)+i\varepsilon_{i}\left(\omega \right)}+1}\right|}^{2}$$
(6)n(ω)=22[εr2(ω)+εi2(ω)+εr(ω)]12

## 3. Results and Discussion

### 3.1. Structural Stability

Figure 1 shows the unit cell, 2 × 2 × 1 supercell and electronic energy band structure of the optimized stable TTG structure. In the unit cell of the TTG sheet, there are two types of inequivalent carbon atoms, namely C1 and C2. C1 atoms are located on the surface planes forming the tetragonal rings, and C2 atoms are found between the planes and join the two tetragonal layers. The optimized lattice constant of the unit cell is 5.493 Å, and the two tetragonal C1 layers are separated by 2.102 Å. The calculated C1–C1, C1–C2, and C2–C2 bond lengths are 1.470, 1.479, and 1.333 Å, respectively. The bond angles of C1–C1–C1, C1–C2–C1, C1–C2–C2, and C1–C1–C2 are 90.00°, 90.54°, 134.73°, and 119.85°, respectively. These results are in good agreement with the values obtained in previous work [1], demonstrating the accuracy of our model structure. Moreover, the calculated energy band gap (1.89 eV) of pristine TTG is consistent with the previously reported TTG band gaps of 1.79 eV and 1.82 eV [1,4]. Thus, TTG is nonmagnetic and semiconducting, and its band gap is much larger than that of twin graphene (0.75 eV) and γ-graphene (0.43 eV) [3,7].

Subsequently, nonmetallic B and P atoms were co-doped into TTG (TTG/BP). First, different doping positions were considered for the co-doped B and P (BP) atoms in the TTG 1 × 1 × 1 unit cell. We studied the structure of TTG doped with one B and one P—that is, (BP)_1_C_46_. The B and P atoms replaced either two C1 atoms, or two C2 atoms, or one C1 atom and one C2 atom in the doped systems, and 13 different configurations were considered, covering all possible B and P positions in the 1 × 1 × 1 unit cell. The optimized doped structures and the corresponding formation energies of the (BP)_1_C_46_ sheets are shown in Figure 2. When P atoms replace C2 atoms (Figure 2f,h,j,l,m), the formation energy of the BP co-doped TTG system is positive, indicating that the co-doped structures are unstable. The formation energy values are all negative for the remaining co-doping systems, indicating that the doping process is exothermic and that the doping systems are thermodynamically stable. This may be because the radius of the P atom (1.1 Å) is larger than those of the C (0.77 Å) and B (0.82 Å) atoms. Using a P atom to replace the C2 atom—which is located midplane and joins the two tetragonal layers—damages the stability and periodicity of the structure. Moreover, it is observed from Figure 2 that the formation energies are lower for the doped configurations (a), (b), (c), (d), (e), with only slight differences between them, indicating that these doping systems are more likely to be prepared experimentally. Studies have shown that the dopants are generally selected to be doped on the same carbon atomic layer for both monolayer carbon allotropes and multilayer carbon allotropes [3,16,17]. Therefore, we further study only the case where two C1 atoms are replaced in the same atomic layer, i.e., the doped configurations shown in Figure 2a,b that appear to be easier to prepare in future experiments and, therefore, are more relevant.

Different doping concentrations were considered in the 2 × 2 × 1 TTG supercell. As shown in Figure 3, the B and P atoms substitute two C1 atoms at the para and ortho positions of the tetragonal C1 layer in 1 × 1 × 1 unit cell; these substitutions are denoted as TTG/BP-para and TTG/BP-ortho systems, respectively. For the different doping concentrations, given the periodicity of the lattice structure, the BP co-doping is considered in the unit cell and in the enlarged 2 × 2 × 1 supercell. We did not consider the co-doping B and P between different unit cells of the 2 × 2 × 1 supercell because such co-doped configurations are quite complicated and are difficult to describe. The TTG/BP systems with different doping concentrations are denoted as TTG/BP-para-4.2%, TTG/BP-para-8.3%, TTG/BP-para-12.5%, and TTG/BP-para-16.7%, and as TTG/BP-ortho-4.2%, TTG/BP-ortho-8.3%, TTG/BP-ortho-12.5%, and TTG/BP-ortho-16.7% systems. The formation energy values (*E_f_*) of the stable co-doped TTG structures are listed in Table 1. The formation energy values are negative for all the co-doping systems, indicating that the doping process is exothermic, and the doping systems are thermodynamically stable. An examination of the data presented in Table 1 shows that for the same doped para or ortho positions, the formation energy decreases with increasing doping concentration, and the stability of the TTG/BP-para or TTG/BP-ortho systems also increases with increasing doping concentration—that is, the symmetry of the TTG/BP systems increases with increasing doping concentration. Consequently, the four-pair BP atom-doped TTG system was the most stable.

For the same doping concentration at different doping positions, the absolute values of the formation energy of the TTG/BP-para system are lower than those of the TTG/BP-ortho system—that is, ortho doping results in more stable systems than para doping. This may be explained by the hybridization of the orbitals between the atoms. The valence electrons configurations for the B, C, and P atoms are 2s^2^2p^1^, 2s^2^2p^2^, and 3s^2^3p^3^, respectively. When the B (2s^2^2p^1^) and P (3s^2^3p^3^) atoms substitute two C1 atoms at the ortho positions of the tetragonal C1 layer, the 2p orbitals of B (2s^2^2p^1^) and P (3s^2^3p^3^) have more unpaired valence electrons than C (2s^2^2p^2^), leading to greater hybridization and stronger interactions between the 2p orbitals of B or P and the 2p orbitals of C.

### 3.2. Electronic Properties

The electron localization function (ELF) diagram was used to characterize the localized distribution of electrons in co-doped TTG systems, as shown in Figure 4a–c. A larger ELF value indicates stronger electron localization while a smaller ELF value indicates stronger delocalization. It is observed that all C–C bonds share the valence electrons of adjacent atoms, demonstrating covalent bond characteristics. Upon B and P co-doping, electron localization of the bonds between the B and P atoms and the surrounding C atoms are enhanced, indicating that the B–C and P–C bonds in the doping system possess some ionic bond characteristics.

Bader charge analysis is the simplest and most direct method for determination of atomic charges and charge transfer analysis. For BP co-doped TTG systems, the calculated charges of B and P are −1.66 *e* and −1.29 *e* in the TTG/BP-para system, and are −1.28 *e* and −0.65 *e* in the TTG/BP-ortho, respectively. The negative charge value represents the charge loss. Consequently, in the doped systems, both B and P atoms lose charge by donating their own charges to the surrounding C atoms of the TTG. This is because the C atom has a higher electronegativity (2.55) than the B (2.04) and P (2.19) atoms. Moreover, charge transfer can also be characterized by the difference charge density, as shown in Figure 4d,e where yellow and blue regions indicate electron gain and loss, respectively. The B and P atoms of all TTG/BP systems are surrounded by blue regions, indicating that B and P atoms are electron donors. It is observed from the electron density that the electron localization of P atoms is stronger in the TTG/BP-ortho system than in the TTG/BP-para system, as was also found from the ELF diagram. In addition, in the TTG/BP-ortho system (Figure 4e), the interaction between the two C atoms around the B and P atoms is enhanced due to the introduction of the B and P dopants.

The band structure of the TTG/BP system is shown in Figure 5. The band gaps of the co-doped systems are listed in Table 1. Regardless of doping, all TTG/BP systems show semiconducting behavior. Compared to the band gap of the pristine TTG, the band gaps of all the co-doped systems are reduced, implying that the transport barrier of the system is reduced and electron transfer is enhanced. The band gaps of the TTG/BP-para and TTG/BP-ortho systems are 1.28 eV and 1.04 eV, respectively, for the 4.2% doping concentration. The band gaps of both the TTG/BP-para and TTG/BP-ortho systems gradually diminish with increasing doping concentration—that is, their electron transfer ability is gradually enhanced. This phenomenon is similar to that observed for the TM-doped TTG systems where higher concentration of the 3d transition metal atoms leads to smaller band gap [4]. Moreover, for the same doping concentration, the TTG/BP-ortho system has a narrower band gap than the TTG/BP-para system. In the TTG/BP-para and TTG/BP-ortho systems, the BP co-doping concentration regulates the TTG band gap. Among all of the doped systems with different BP concentrations, the TTG/BP-ortho-16.7% system has the smallest band gap (0.70 eV). The band gaps of all doped systems are in the range of 0.7~1.28 eV. These band gaps are very close to that of the widely used silicon semiconductor material (1.16 eV), suggesting that these doped systems have great potential for applications in semiconductor devices such as pn junction diodes, metal oxide field-effect transistors (MOSs), bipolar transistors (BJTs), and junction field-effect transistors (JFETs).

Next, we discuss the partial density of states (PDOS) of the co-doped TTG systems shown in Figure 6. The valence band maximum (VBM) of the TTG/BP-para doping systems with different concentrations are formed by the hybridization of the C-*p* (the sum of C-*p_x_*, C-*p_y_*, and C-*p_z_* orbitals), B-*p_y_*, P-*p_x_*, and P-*p_y_* orbitals. The conduction band minimum (CBM) is formed by the hybridization of C-*p_z_*, B-*s*, B-*p_y_*, and P-*p_z_* orbitals. With increased doping concentration, the contribution of the C-*p_x_* orbital gradually increased relative to those of the C-*p_y_* and C-*p_z_* orbitals and the VBM formed by the hybridization of the C-*p_x_* orbitals and other orbitals (B-*p_y_*, P-*p_x_*, P-*p_y_*) shifted upward. Combining these findings with the results presented in Figure 5, we find that the energy contributions of different atoms near the Fermi level are different, resulting in the reduction of the band gap. For all TTG/BP-ortho doping systems, the hybridization of the C-*p*, B-*p_y_*, and P-*p_y_* orbitals contributed to the VBM, and the hybridization of the C-*p*_z_, B-*s*, B-*p_y_*, P-*p_z_* orbitals contributed to the CBM. With increased doping concentration, the VBM and CBM of the TTG/BP-ortho doped systems gradually move toward the Fermi level, resulting in the reduction of the band gap in the TTG/BP-ortho doped systems. In addition, the coupling effect between the C, B, and P atoms near the Fermi level is enhanced by increasing BP doping concentration. These results indicate that the doping concentration of the TTG/BP systems can strongly regulate the electronic properties, showing potential for the application of the BP-doped TTG systems in nanoelectronics.

### 3.3. Optical Properties

Optical properties are another important feature of two-dimensional materials and particularly of graphene. Therefore, we investigated the absorption coefficient (α(ω)), reflectance (r(ω)), and refractive index (n(ω)) of the TTG/BP systems.

The optical absorption spectrum is the most important optical parameter; for the doping concentration of 4.2%, Figure 7 shows the absorption coefficients (α(ω)) of the pristine TTG, TTG/BP-para, and TTG/BP-ortho systems parallel ($\alpha^{xx}$, $\alpha^{yy}$) and perpendicular ($\alpha^{zz}$) to the TTG surface. A magnification of the region between 0.8 and 2.4 eV is shown in the inset in the top right corner of Figure 7. It is observed that the optical band gap of TTG is approximately 2.06 eV, which is much larger than the 0.75 eV of monolayer graphene [38]. As shown in Figure 7, the optical band gap is the value of the intersection between the reverse tangent and the *x*-axis. For the TTG system, the adsorption coefficients for the parallel components $\alpha^{xx}$ and $\;\alpha^{yy}$ are the same, and $\alpha^{xx}$ and $\alpha^{yy}$ are different from the vertical component $\;\alpha^{zz}$. Moreover, the adsorption coefficients of the TTG systems are clearly different from those of the monolayer and bilayer graphene [38,39]—that is, the optical properties of TTG are different from those of graphene.

After BP co-doping, the $\alpha^{xx}$, $\alpha^{yy}$, and $\alpha^{zz}$ adsorption coefficients are markedly different, indicating significant optical anisotropy. All of the main absorption peaks are found in the ultraviolet region, and the peaks of the TTG/BP systems are significantly reduced at 5, 10, and 15 eV compared to the TTG system. The BP co-doping changes the maximum absorption peak from the vacuum ultraviolet region (10.5 eV) of the pristine TTG to the near-ultraviolet regions (4.8 eV) of the TTG/BP systems for the parallel components $\alpha^{xx}$ and $\alpha^{yy}$, but for the vertical component $\alpha^{zz}$, the main absorption peak is still located at 9.8 eV in the vacuum ultraviolet region. In the low-energy (1.6~2.2 eV) zone of the visible region (1.6~3.2 eV), the BP co-doping increases the absorption coefficient, and the enhancement of the absorption coefficient of the TTG/BP-ortho system is larger than that of the TTG/BP-para system for $\alpha^{xx}$ and $\alpha^{yy}$, whereas the opposite is true for $\alpha^{zz}$. A similar phenomenon occurs for $\alpha^{zz}$ in the near-infrared region (1.2~1.6 eV). For all TTG and TTG/BP systems, the absorption edge appears in the mid-infrared region (approximately 1.2 eV), except for the presence of a small peak for the $\alpha^{yy}$ of the TTG/BP-ortho system. The optical band gaps are 1.09 and 1.30 eV for the TTG/BP-para and TTG/BP-ortho systems, respectively, which are smaller than that of pristine TTG (2.06 eV) because of the intraband transition. These results are consistent with the decrease in the energy band gap upon BP co-doping. Consequently, a lower energy can be used for the photoexcitation of electrons in BP co-doped TTG systems because of their smaller optical band gap. In addition, the absorption edge of TTG/BP system is redshifted, implying an increased utilization rate of the visible light for this system. This phenomenon is related to the reduction of the band gap caused by the co-doping of BP.

The effect of doping concentration on the absorption coefficient of TTG/BP systems is shown in Figure 8. In the near-ultraviolet region (3.2~5.0 eV), the absorption coefficient monotonically decreases with increasing doping concentration, except for the $\alpha^{yy}$ of the TTG/BP-para system. Conversely, in the vacuum ultraviolet region (6.0~13.0 eV), the absorption coefficient monotonically increased with increasing doping concentration. Moreover, the absorption peaks for $\alpha^{zz}$ are redshifted from the near-ultraviolet region to the visible region (1.6~3.2 eV) for both the TTG/BP-para and TTG/BP-ortho systems. Thus, BP doping concentration primarily affects the absorption coefficient in the visible region and the vacuum ultraviolet region. These results show that increased utilization of visible light is obtained for TTG by BP co-doping. Moreover, for the TTG/BP-para systems, the doping concentration has different effects on the absorption coefficient in different directions ($\alpha^{xx}$, $\alpha^{yy}$, and $\;\alpha^{zz}$), and the maximum absorption peaks in the *x*, *y*, and *z-* directions are obtained by 4.2%, 8.3%, and 16.7% doping, respectively.

Figure 9 shows the reflectance $r^{xx}$, $r^{yy}$, and $r^{zz}$ of the pristine TTG and TTG/BP systems in the *x*-, *y*-, and *z*-directions, respectively. It is observed that the reflectance spectrum of TTG differs from that of graphene [38], and the main reflectance peaks are found in the near-ultraviolet region (3.2~5.0 eV), revealing that the TTG and TTG/BP systems can be possibly used for short-wavelength optoelectronic devices. In addition, for a given doping concentration, the values of the main reflectance peaks of the TTG/BP-para system are larger than those of the TTG/BP-ortho system. In the infrared region (0~1.6 eV), reflectance gradually increases with increasing doping concentration, and large peaks are produced for all BP co-doped TTG systems, although $r^{zz}$ is essentially unchanged in the TTG/BP-ortho system. In the visible region (1.6~3.2 eV), the maximum doping concentration (16.7%) induces the maximum reflectance, with no obvious relationship between the reflectance and doping concentration, with the exception of $r^{zz}$ that gradually decreases with increasing doping concentration. In the ultraviolet region (3.2~13.0 eV), the reflectance of the TTG/BP systems decreases dramatically with an increasing doping concentration at 4.8 eV for $r^{xx}\;$ and $r^{yy}$, and at 3.4 eV for $r^{zz}$. Meanwhile, for $r^{zz}$ of the TTG/BP systems, the reflectance peaks are redshifted from the near-ultraviolet region (3.4 eV) to the visible region (1.6~3.2 eV) with increasing doping concentration. Moreover, BP co-doping causes the two peaks to disappear at approximately 10.0 eV in the vacuum ultraviolet region. These results show that light transmittance in TTG is enhanced after BP doping—that is, BP-doped TTG has potential application in optical waveguide devices.

Refractive index is another important optical property of materials; Figure 10 shows the refractive indices $n^{xx}$, $n^{yy}$, and $n^{zz}$ of pristine TTG and of TTG/BP systems in the *x*-, *y*-, and *z*-directions, respectively. The refractive spectrum of TTG is similar to that of graphene [38], with the peak refractive index occurring in the 0~5.0 eV region. Similar to TTG reflectance, the refractive index in the infrared region (0–1.6 eV) gradually increases with increasing doping concentration, and significant peaks are produced for all BP co-doped TTG systems. However, an interesting phenomenon is observed based on the comparison between the results for the two doping systems shown in Figure 10c,f,i. For the vertical component $n^{zz}$, para and ortho site doping have completely different effects, with the refractive index remaining essentially unchanged for the TTG/BP-ortho system. The visible region (1.6~3.2 eV) has the largest refractive index, indicating strong refraction by the pristine TTG and TTG/BP systems in this region of the spectrum. Most importantly, comparison of the three refractive indices shows that the maximum refractive index is obtained in the z-direction, and $\;\;n^{zz}$ gradually decreases with increasing doping concentration. Consequently, the TTG and BP co-doped TTG systems can be used as reflectors with a high refractive index (2.0~2.5), and as dielectric filters with high transmittance in the visible range. In the ultraviolet region (3.2~13.0 eV), a minimum value of the refractive index and maximum reflectance are obtained at 4.9 eV, as shown in Figure 9. In particular, for $n^{zz}$, the refractive index at 3.9 eV increases significantly with increasing doping concentration in the near-ultraviolet region. In conclusion, the values of the absorption coefficient, reflectance, and refractive index in the ultraviolet region decrease—that is, BP co-doping improves the transmittance of TTG. In the infrared and visible regions, the increased reflectivity and refraction and the reduced absorption coefficient make BP co-doped TTG systems suitable for use in optical communication devices.

## 4. Conclusions

This study used first-principles calculations to systematically examine the structural stability and the electronic and optical properties of the TTG/BP systems formed by the co-doping of TTG with B and P atoms. The stability of the TTG/BP systems increased with increasing doping concentration. The position and concentration of BP co-doping can regulate the TTG band gap. All TTG/BP systems remain semiconductors with band gaps that gradually decrease with increasing doping concentration. TTG and the BP co-doped TTG systems exhibited significant optical anisotropy. The absorption coefficient, reflectance, and refractive index values decreased in the ultraviolet region upon co-doping, indicating that B and P co-doping improves the transmittance of TTG. The increased reflectivity and refraction, and the reduced absorption coefficient in the infrared and visible regions, make BP co-doped TTG suitable for use in optoelectronic devices.

## Figures and Tables

**Figure 1 materials-15-02876-f001:**
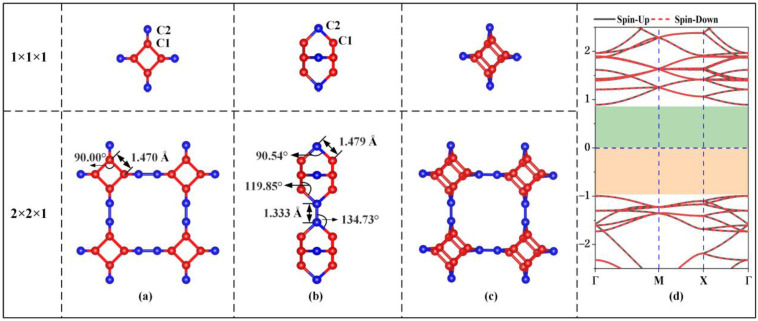
Geometrical structure (**a**) Top view; (**b**) Side view; (**c**) Bird’s-eye view, and electronic band structure (**d**) of TTG.

**Figure 2 materials-15-02876-f002:**
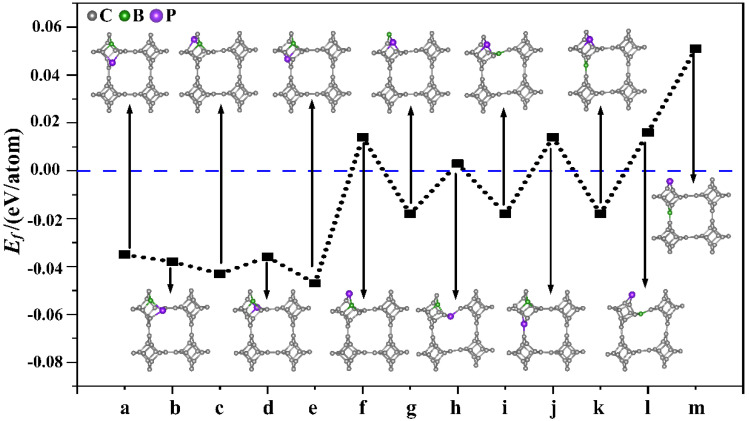
Different doping systems (TTG/BP) and their corresponding formation energies. (**a**–**m**) are 13 different doping configurations.

**Figure 3 materials-15-02876-f003:**
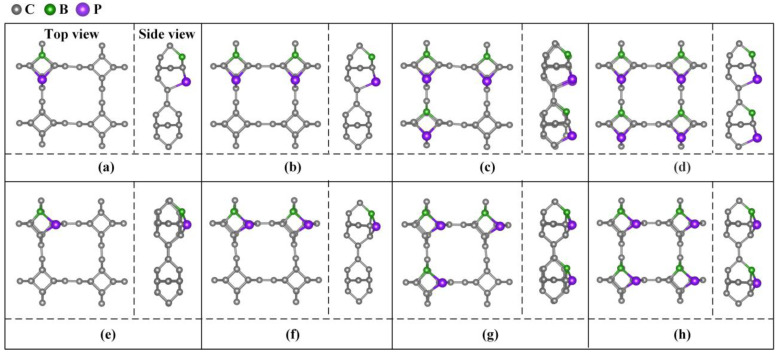
B and P co-doped 2 × 2 × 1 TTG supercell with doping at the para (TTG/BP-para) and ortho (TTG/BP-ortho) positions with different doping concentrations. (**a**) TTG/BP-para-4.2%; (**b**) TTG/BP-para-8.3%; (**c**) TTG/BP-para-12.5%; (**d**) TTG/BP-para-16.7%; (**e**) TTG/BP-ortho-4.2%; (**f**) TTG/BP-ortho-8.3%; (**g**) TTG/BP-ortho-12.5%; (**h**) TTG/BP-ortho-16.7%.

**Figure 4 materials-15-02876-f004:**
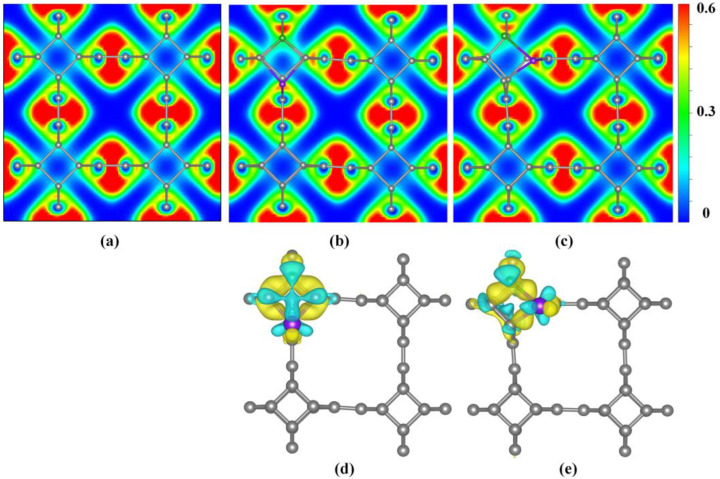
Electron localization function (ELF) (**a**) TTG; (**b**) TTG/BP-para; (**c**) TTG/BP-ortho; and difference charge density (**d**) TTG/BP-para; (**e**) TTG/BP-ortho of pristine TTG and co-doped TTG systems.

**Figure 5 materials-15-02876-f005:**
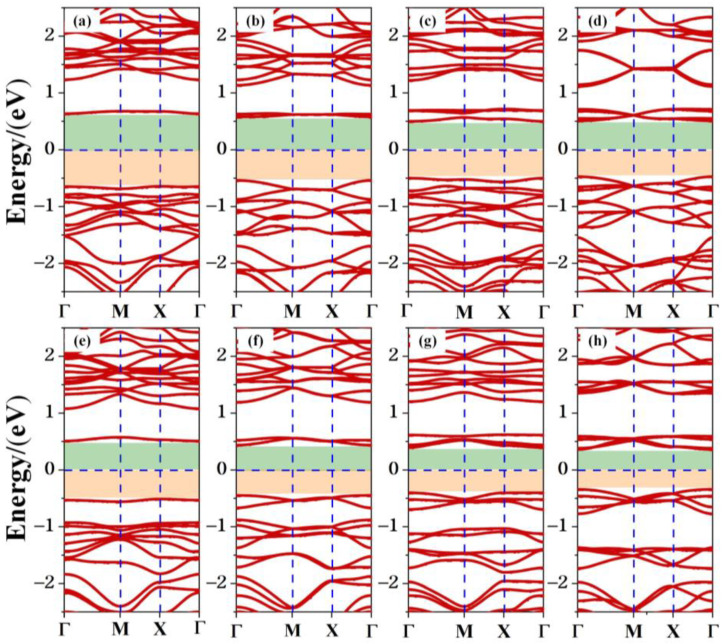
Electronic band structure of co-doped TTG systems. (**a**) TTG/BP-para-4.2%; (**b**) TTG/BP-para-8.3%; (**c**) TTG/BP-para-12.5%; (**d**) TTG/BP-para-16.7%; (**e**) TTG/BP-ortho-4.2%; (**f**) TTG/BP-ortho-8.3%; (**g**) TTG/BP-ortho-12.5%; (**h**) TTG/BP-ortho-16.7%.

**Figure 6 materials-15-02876-f006:**
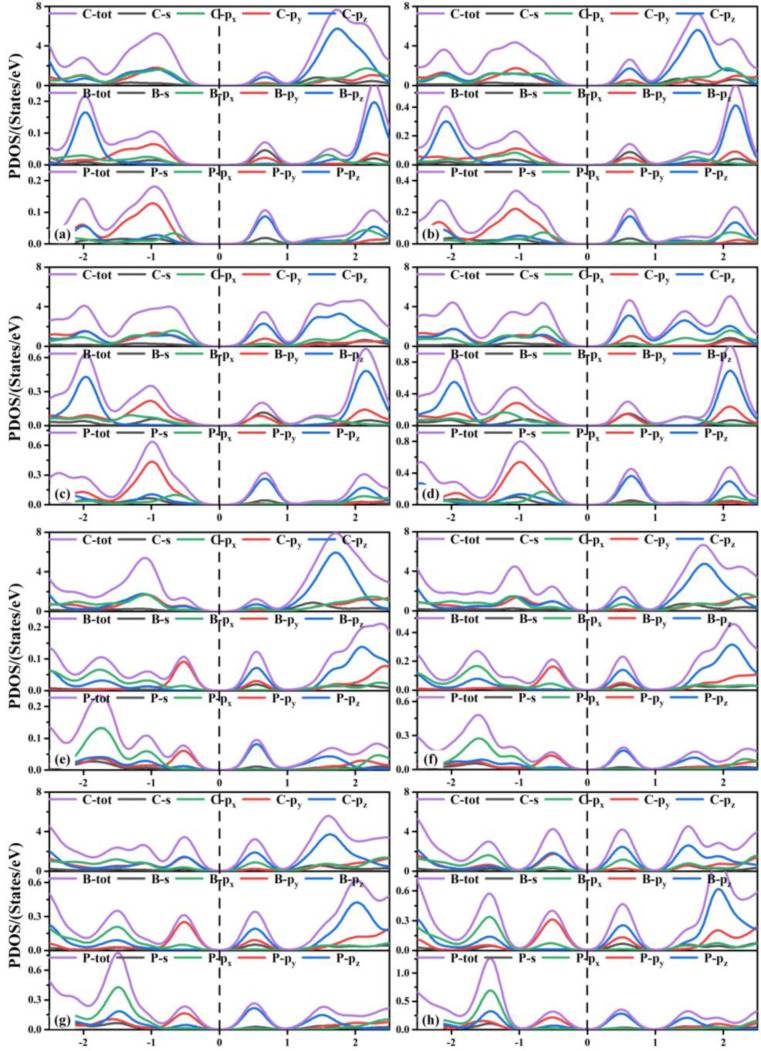
Partial density of states (PDOS) of TTG/BP systems. (**a**) TTG/BP-para-4.2%; (**b**) TTG/BP-para-8.3%; (**c**) TTG/BP-para-12.5%; (**d**) TTG/BP-para-16.7%; (**e**) TTG/BP-ortho-4.2%; (**f**) TTG/BP-ortho-8.3%; (**g**) TTG/BP-ortho-12.5%; (**h**) TTG/BP-ortho-16.7%.

**Figure 7 materials-15-02876-f007:**
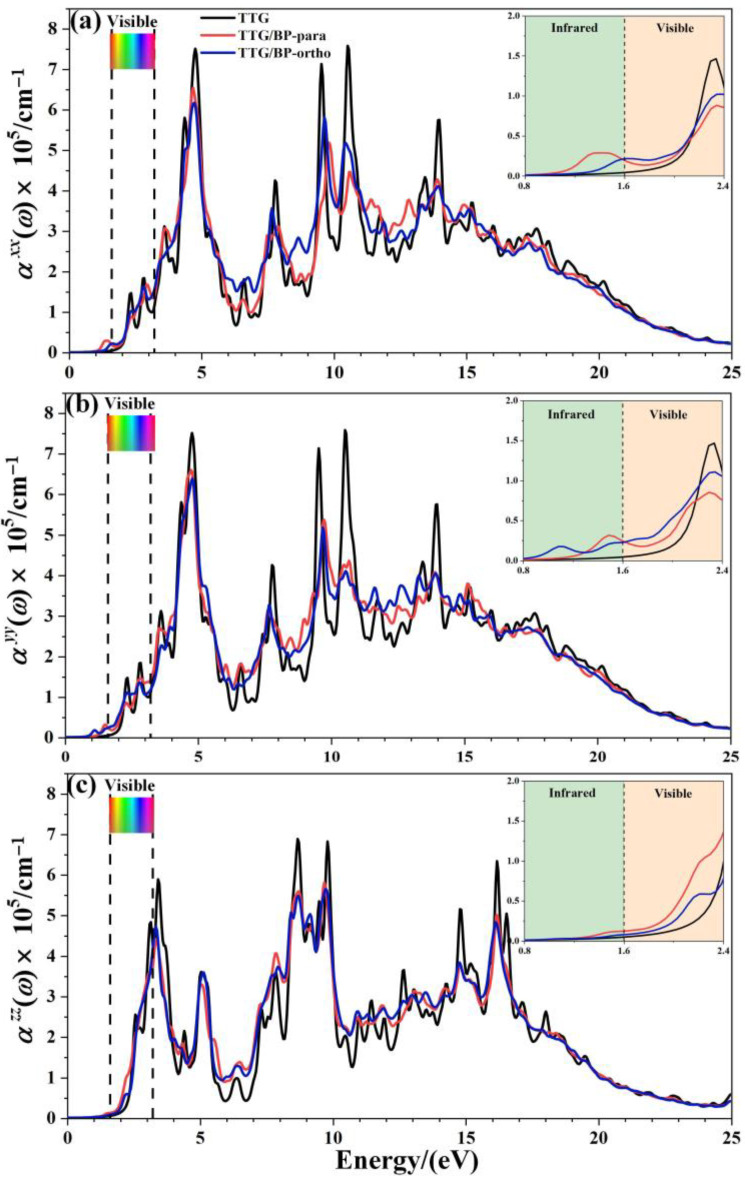
Absorption coefficient of pristine TTG and TTG/BP-4.2% systems. (**a**) $\alpha^{xx}$ of TTG/BP-para-4.2%; (**b**) $\alpha^{yy}$ of TTG/BP-para-4.2%; (**c**) $\alpha^{zz}$ of TTG/BP-para-4.2%.

**Figure 8 materials-15-02876-f008:**
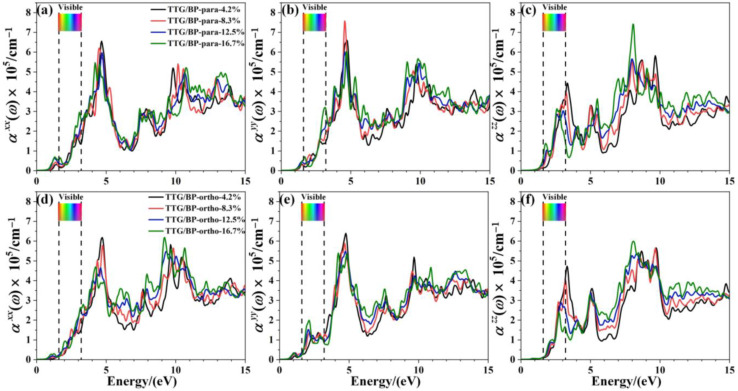
Absorption coefficient of TTG/BP system with different doping concentrations. (**a**) $\alpha^{xx}$ of TTG/BP-para; (**b**) $\alpha^{yy}$ of TTG/BP-para; (**c**) $\alpha^{zz}$ of TTG/BP-para; (**d**) $\alpha^{xx}$ of TTG/BP-ortho; (**e**) $\alpha^{yy}$ of TTG/BP-ortho; (**f**) $\alpha^{zz}$ of TTG/BP-ortho.

**Figure 9 materials-15-02876-f009:**
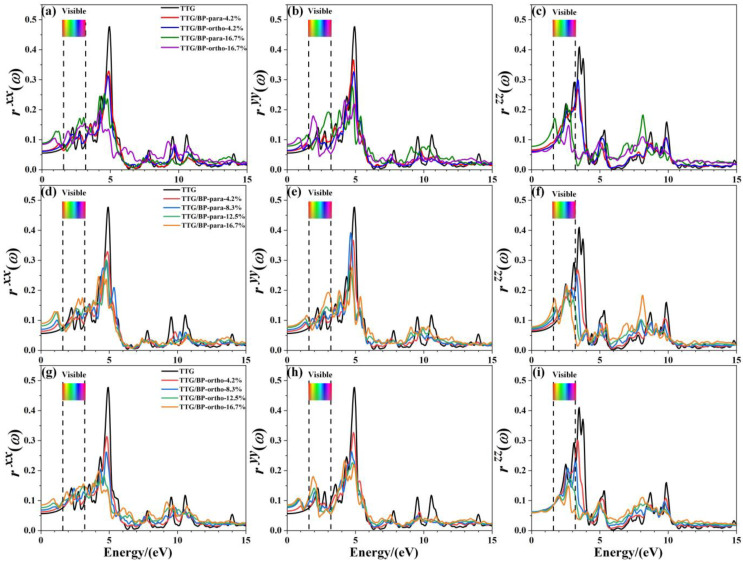
Reflectance of the TTG/BP systems. (**a**) $r^{xx}$ of TTG/BP; (**b**) $r^{yy}$ of TTG/BP; (**c**) $r^{zz}$ of TTG/BP; (**d**) $r^{xx}$ of TTG/BP-para; (**e**) $r^{yy}$ of TTG/BP-para; (**f**) $r^{zz}$ of TTG/BP- para; (**g**) $r^{xx}$ of TTG/BP-ortho; (**h**) $r^{yy}$ of TTG/BP-ortho; (**i**) $r^{zz}$ of TTG/BP-ortho.

**Figure 10 materials-15-02876-f010:**
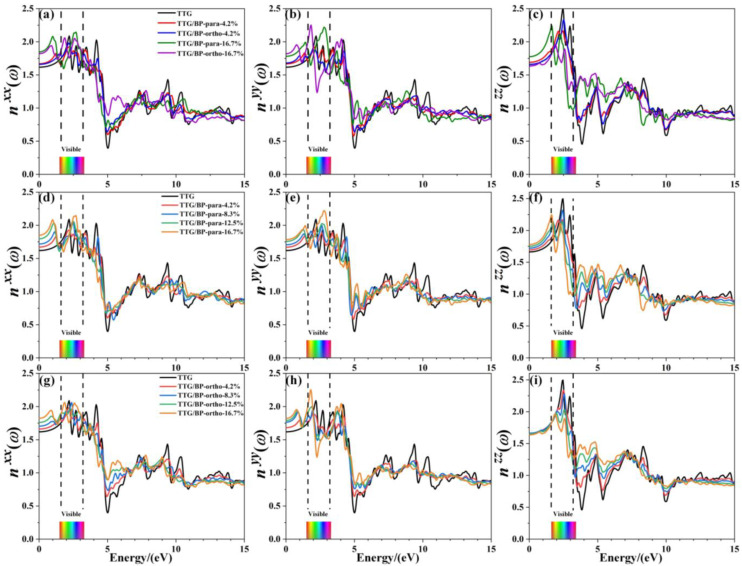
Refractive index of TTG/BP systems. (**a**) $n^{xx}$ of TTG/BP; (**b**) $n^{yy}$ of TTG/BP; (**c**) $n^{zz}$ of TTG/BP; (**d**) $n^{xx}$ of TTG/BP-para; (**e**)$\;n^{yy}$ of TTG/BP- para; (**f**)$n^{zz}$ of TTG/BP- para; (**g**) $n^{xx}$ of TTG/BP-ortho; (**h**) $n^{yy}$ of TTG/BP-ortho; (**i**)$n^{zz}$ of TTG/BP-ortho.

**Table 1 materials-15-02876-t001:** Formation energy (*E_f_*) and band gap values of co-doped TTG systems.

Position	Concentration (%)	*E_f_* (eV)	Band Gap (eV)
Para	4.2	−0.035	1.28
	8.3	−0.069	1.12
	12.5	−0.100	1.00
	16.7	−0.131	0.98
Ortho	4.2	−0.038	1.04
	8.3	−0.075	0.86
	12.5	–0.112	0.80
	16.7	−0.148	0.70

## Data Availability

Not applicable.

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
