# Peer review of "Electronic and Optical Properties of Twin T-Graphene Co-Doped with Boron and Phosphorus"

_materials, 2022, doi:10.3390/ma15082876_

Round 1

Reviewer 1 Report

This paper analyzes the influence of co-doping on the properties of twin T‐graphene. The topic is interesting, and I believe that there are some few points that could addressed by the authors:

  1. 13 different configurations of B and P atoms were considered. Does this number of configurations cover all the possibilities for the B and P positions? And what about B replacing C2 atoms?
  2. Is it feasible to fabricate any of these stable configurations? Or is any of them most probable of being obtained? Can the positions of the B and P atoms be controlled during fabrication process?
  3. I could not understand what the authors mean when stating: “The charge transfer was calculated qualitatively”. If the quantity is calculated, I believe that analysis would be quantitative. Or do the calculated values have no direct relation to the quantity of charges?

Reviewer 2 Report

Reviewer Comment for Author

Manuscript ID: materials-1628294

Titled:  Electronic and optical properties of twin T-graphene co-doped with boron and phosphide

Journal: materials

Recommendation:

The manuscript presents interesting achievements in the field. It may be recommended to be published in the materials Journal. However, the paper is not well written and the paper has many weaknesses. The idea and the results need more depth.

  1. The novelty of the work should be highlighted to real physics phenomena in both the introduction and abstract. This kind of problem has been studied many times and in the same way.
  2. The mathematical model is unclear as well as the solution method.
  3. The discussion is sketchy. It is more graphical presentation of various numerical computation, lacks depth and physical contents. The most important results obtained should be clearly highlighted.
  4. Extensive editing of English language and style is required. Numerous grammatical and spelling errors greatly degraded the quality of the presentation.
  5. There exist some un-defined acronyms through the text, which need to be properly declared.
  6. In the discussion section, why was there no comparison between the different models, especially since the model used has flaws like the classic model of heat transport equation? You can use https://doi.org/10.3390/math9222902 
  7. Write the conclusion more precious.

Reviewer 3 Report

The paper is really interesting, the authors present the simulation of the optoelectronic properties of a TTG, a new 2D material. I only miss a concrete applicability for this material. PLease, describe more strongly  in the introduction section for which devices and what would be the reasons for which this material could be useful.

And if it would be possible, I would like to see not only the design but an experimental work. 

Round 2

Reviewer 1 Report

I believe the authors have adequately answered my comments on the previous manuscript version. Please, provide a reference for eqs. (2)-(5). There are just a few typos:

  • Line 33 – allotrope of with a thickness
  • Line 39 – 375 cm2∙V–1∙S–1
  • Line 153 – covering all possible of B and P positions
  • Line 247 – partial density state (PDOS)

Author Response

Thank you for your careful examination and we are very sorry for our carelessness. We have corrected the spelling mistakes and provide the relevant reference for eqs. (2)-(5) in the revised manuscript as following:

  • Line 33 – allotrope with a thickness
  • Line 39 – 375 cm2·V–1·s–1
  • Line 153 – covering all possible B and P positions
  • Line 247 – partial density of state (PDOS)

[38] Qiu, B.; Zhao, X.; Hu, G.; Yue, W.; Ren, J.; Yuan, X., Optical properties of graphene/MoS2 heterostructure: First principles calculations. Nanomaterials 2018, 8, (11), 962. 

Reviewer 2 Report

After reviewing the revised version, it was found that the authors had made the required improvements. They also responded to some concerns. I think now the manuscript is publishable.

Author Response

Thank you very much. Your comments and suggestions have greatly improved the quality of our articles.
